# Static magnetic field stimulation of the supplementary motor area modulates resting-state activity and motor behavior

José A. Pineda-Pardo [1,6], Ignacio Obeso[1,6], Pasqualina Guida[1], Michele Dileone[1], Bryan A. Strange [2,3], José A. Obeso[1,4], Antonio Oliviero[5] & Guglielmo Foffani [1,5]*

Focal application of a strong static magnetic field over the human scalp induces measurable local changes in brain function. Whether it also induces distant effects across the brain and how these local and distant effects collectively affect motor behavior remains unclear. Here we applied transcranial static magnetic field stimulation (tSMS) over the supplementary motor area (SMA) in healthy subjects. At a behavioral level, tSMS increased the time to initiate movement while decreasing errors in choice reaction-time tasks. At a functional level, tSMS increased SMA resting-state fMRI activity and bilateral functional connectivity between the SMA and both the paracentral lobule and the lateral frontotemporal cortex, including the inferior frontal gyrus. These results suggest that tSMS over the SMA can induce behavioral aftereffects associated with modulation of both local and distant functionally-connected cortical circuits involved in the control of speed-accuracy tradeoffs, thus offering a promising protocol for cognitive and clinical research.

[1] CINAC, Hospital Universitario HM Puerta del Sur, Móstoles, Universidad CEU-San Pablo, Madrid, Spain. [2] Laboratory for Clinical Neuroscience, CTB, Universidad Politecnica de Madrid, Madrid, Spain. [3] Department of Neuroimaging, Alzheimer's Disease Research Centre, Reina Sofia-CIEN Foundation, Madrid, Spain. [4] CIBERNED, Instituto de Salud Carlos III, Madrid, Spain. [5] Hospital Nacional de Parapléjicos, Toledo, Spain. [6] These authors contributed equally: José A. Pineda-Pardo, Ignacio Obeso. *email: gfoffani.hmcinac@hmhospitales.com

The supplementary motor area (SMA), located in the medial frontal cortex, delimits the motor from the prefrontal cortical areas and is critical for linking cognition to action in normal behaviors[1]. The most posterior section of the SMA, the SMA proper[2], is directly connected to the primary motor cortex (M1) with scarce prefrontal projections[3] and has been traditionally associated to programming and anticipation of motor plans[4–7]. In contrast, the most anterior section of the SMA—the pre-SMA[2] has wide connections to the prefrontal cortex[3,8] and is predominantly involved in cognitive operations underlying behavioral change, such as action switching[9,10], movement stopping[11–13], and setting speed-accuracy tradeoffs in decision making[14,15]. This unique role played by the SMA as a crossroads between cognition and action makes it an attractive non-invasive brain stimulation (NIBS) target for treating neurological and psychiatric disorders in which the control of movements or behaviors is altered, including Tourette syndrome[16–18], obsessive compulsive disorder[19], and Parkinson's disease[20,21].

Transcranial static magnetic field stimulation (tSMS) is a recent NIBS technique that consists of focally applying a relatively strong neodymium magnet over the scalp[22]. tSMS induces a reduction of cortical excitability that outlasts the duration of the stimulation, as measured by decreased motor-evoked potentials, somatosensory-evoked potentials, and intra-epidermal electrical stimulation-evoked potentials when applied over the motor cortex[22–32], and by locally increased electroencephalography α-activity when applied over the visual cortex[33] or the somatosensory cortex[34]. Despite one study reporting negative findings[35], possibly due to methodological differences[36], converging evidence supports the ability of tSMS to induce local effects. Conversely, little is known about the ability of tSMS to induce neuroanatomically distant effects across the brain[37] and to affect motor behavior[38–40].

Here we applied tSMS over the SMA (i.e. both SMA proper and pre-SMA) in healthy subjects (Fig. 1a), running two independent experiments in order to establish the behavioral relevance and uncover the network effects of the intervention. Experiment 1 was a randomized double-blind sham-controlled parallel study in which we measured the behavioral after-effects of tSMS (Fig. 1b; Experiment 1a in Results and Methods section). We used three different choice-reaction time (CRT) tasks to assess the impact of SMA tSMS on withholding predicted actions (fully-cued), motor planning (uncued-compatible) and stimulus–response compatibility (uncued-incompatible) during motor choice behavior, and we fitted drift-diffusion models to the data in order to gain insight about the underlying cognitive processes[41]. A subgroup of subjects also repeated the experiment in a crossover design (Experiment 1b) and a new group performed the same

experiment but with tSMS of M1 (Experiment 1c). Experiment 2 was a randomized double-blind sham-controlled crossover study in which we measured the functional after-effects of SMA tSMS using resting-state functional magnetic resonance image (fMRI) (Fig. 1c). We investigated the local and distant tSMS after-effects by quantifying the amplitude of low-frequency fluctuations (ALFF), the regional homogeneity (ReHo), and the whole-brain seed-based functional connectivity using left and right SMA regions of interest (ROIs).

## Results

**Experiment 1a: Behavioral findings, parallel design**. We tested the behavioral after-effects of 30-min tSMS of the SMA in a double-blind sham-controlled parallel study on 42 healthy subjects who performed three CRT tasks (Fig. 2a): 20 subjects performed the tasks after receiving real tSMS (mean ± SD; 31.9 ± 9.3 years old, 14 females) and 22 after receiving sham tSMS (31.2 ± 8.4, 14 females). Real and sham groups were matched for age (unpaired $t$ test: $p = 0.82$) and gender (two proportion $z$ test: $p = 0.66$). One subject who received sham tSMS did not complete the fully-cued task and another subject who received sham tSMS did not complete the uncued-compatible task, because they did not follow correctly the instructions.

Reaction times of correct trials were progressively longer, as expected, from the fully-cued (575.5 ± 102.9 ms), to the uncued-compatible (688.1 ± 118.2 ms), to the uncued-incompatible tasks (817.7 ± 152.7 ms; three-way independent-measures analysis of variance (ANOVA), Task: $F_{(2,472)} = 144.4$, $p < 0.0001$). Importantly, reaction times were longer in subjects that received tSMS (709.4 ± 166.1 ms) compared to sham (681.0 ± 150.8 ms; three-way independent-measures ANOVA, Stimulation: $F_{(1,472)} = 6.8$, $p = 0.0096$), independently of the task (Stimulation × Task: $F_{(2,472)} = 0.1$, $p = 0.87$) or the target location (Stimulation × Location: $F_{(3,472)} < 0.1$, $p > 0.99$; Stimulation × Task × Location: $F_{(6,472)} < 0.1$, $p > 0.99$). The longer reaction times after tSMS were mostly due to longer initiation times (Intervention: $F_{(1,472)} = 8.3$, $p = 0.0041$), with little difference of movement times (Stimulation: $F_{(1,472)} = 2.6$, $p = 0.11$; Fig. 2b).

The overall error rate was low: 3.1% of trials in the fully-cued task, due to anticipation errors, 0.7% in the uncued-compatible task, and 1.9% in the uncued-incompatible task (three-way independent-measures ANOVA, Task: $F_{(2,472)} = 19.3$, $p < 0.0001$). Nevertheless, the error rate was lower in subjects that received tSMS compared to sham (Stimulation: $F_{(1,472)} = 9.8$, $p = 0.0018$), depending on the task (Stimulation × Task: $F_{(2,472)} = 4.6$, $p = 0.0107$). Specifically, in the fully-cued task the error rate was lower after tSMS compared to sham (follow-up two-way ANOVA, Stimulation: $F_{(1,156)} = 9.7$,

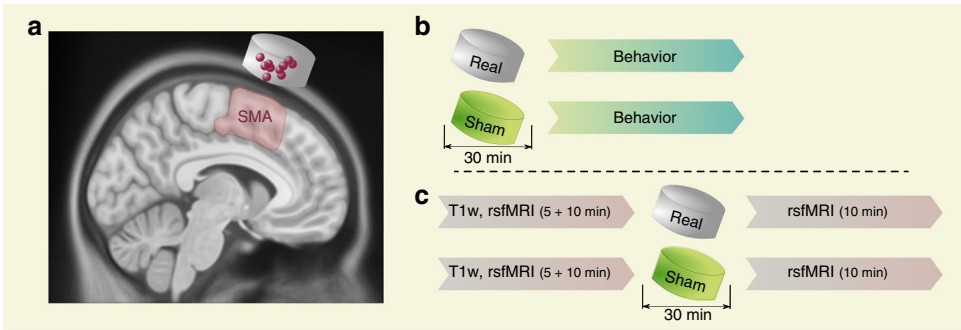

**Fig. 1** Experimental procedure. **a** Transcranial static magnetic field stimulation (tSMS) applied over the supplementary motor area (SMA). The image is a T1-weighted (T1w) magnetic resonance image (MRI) in standard space, with a cartoon magnet/sham centered over the average SMA target (3 cm anterior to Cz) in 10 representative subjects (dots), as confirmed by neuronavigation. **b** Behavioral protocol (behavior refers to choice-reaction time tasks, see Fig. 2). **c** Resting-state functional MRI (rsfMRI) protocol

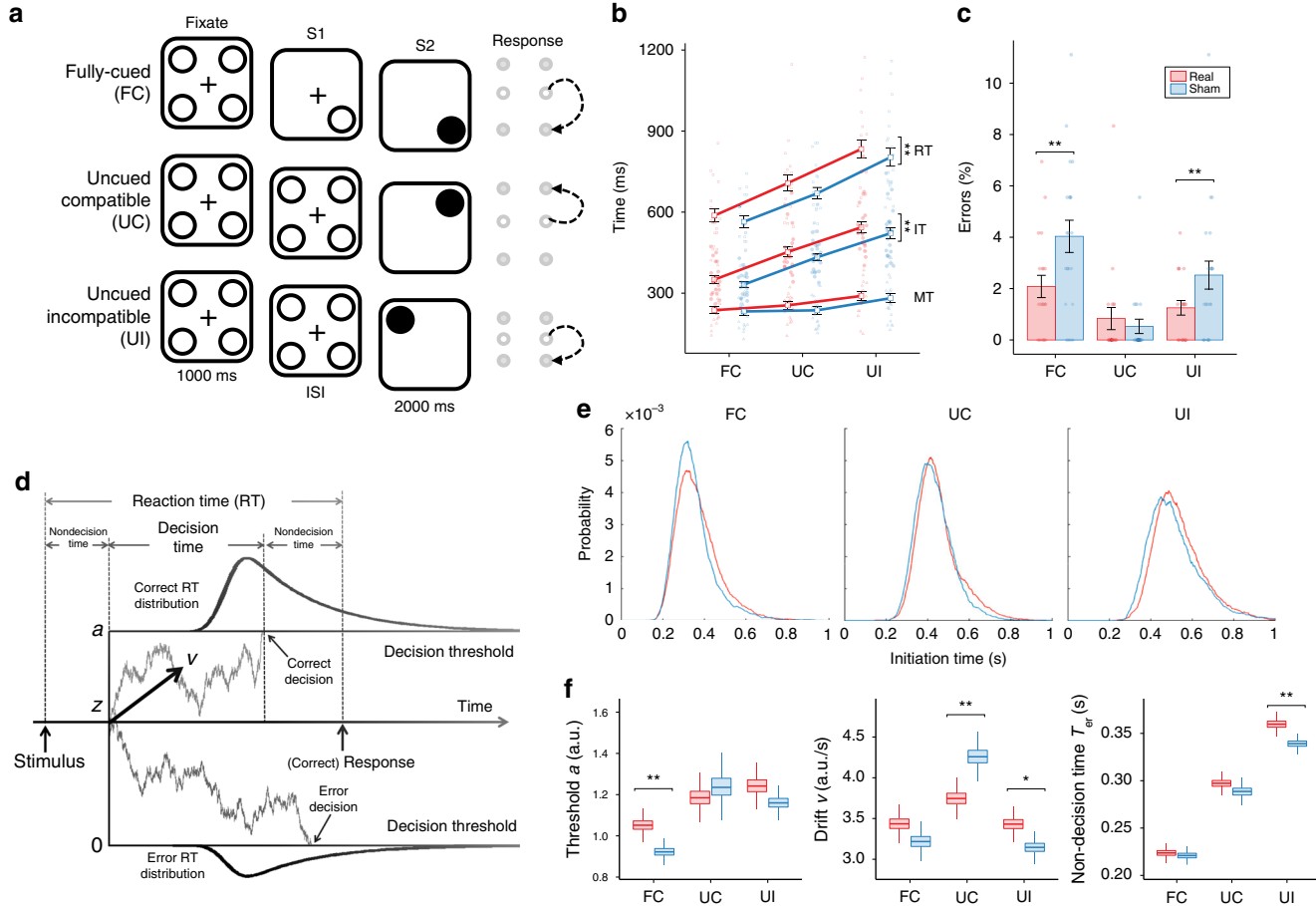

**Fig. 2** Behavioral effects induced by tSMS of the SMA. **a** Schematic representation of the choice-reaction time tasks. All tasks start with a fixation screen with four empty circles, followed by a cue screen (S1) and a GO screen (S2). In the fully-cued (FC) task, the cue is an empty circle in one position (bottom right in the example) representing the target button, and the GO is a filled circle in the same position, triggering the response. In the uncued-compatible (UC) task, the cue screen is equal to the fixation screen, so no target information is provided before the go signal (top right in the example). The uncued-incompatible (UI) task is identical to the UC task, but the subject is instructed to respond to the target that is opposite to the one indicated by the go signal (in the example the go signal indicates top left and the subject needs to respond to bottom right). **b** Average reaction times (RTs), initiation times (ITs), and movement times (MTs) for the three tasks after real or sham tSMS. **c** Corresponding error rates. Error bars represent standard errors ($n_{real} = 20$ and $n_{sham} = 22$). *$p < 0.05$, **$p < 0.01$. **d** Schematic representation of the drift-diffusion model (modified from Fig. 16 by Murata et al.[152], used under CC BY 4.0). Reaction times are modeled as generated by a noisy process that integrates evidence with a positive drift rate $v$ from a starting point $z$ ($z = a/2$ in the EZ-diffusion model) until reaching either a higher threshold $a$, leading to a correct decision, or a lower threshold 0 (i.e. zero), leading to an error decision. The non-decision time $T_{er}$ includes both perceptual processes occurring before decision onset and motor processes occurring after the decision. The EZ-diffusion model estimates $a$, $v$, and $T_{er}$ from the overall probability of correct responses, the mean, and the variance of the reaction times. **e** Distributions of initiation times pooled across all subjects separately for the three tasks and for the groups that received tSMS (red) or sham stimulation (blue). Distributions were smoothed for illustration purposes. **f** Corresponding parameters estimated with the EZ-diffusion model, using bootstrapping techniques. On each box, the central mark represents the median, the edges of the box are the 25th and 75th percentiles, and the whiskers extend to the largest and smallest values within 1.5 times the inter-quartile range from the edges. *$P < 0.05$, **$P < 0.01$. Data for Fig. 2 are provided at https://osf.io/n3au4/

$p = 0.0021$), no differences were observed in the uncued-compatible task ($F(1,156) = 0.7$, $p = 0.39$), and in the uncued-incompatible task the error rate was again lower after tSMS compared to sham ($F(1,160) = 2.7$, $p = 0.0451$) (Fig. 2c).

At the end of the experiment, there was a tendency for subjects to correctly guess whether they had received real tSMS (14 of 20 correct guesses) or sham (13 of 22 correct guesses; $\chi^2 = 3.58$, $p = 0.0585$). Interestingly, correctly guessing real tSMS—but not sham—was associated with slower initiation times (four-way independent-measures ANOVA, Stimulation × Guessing: $F(1,448) = 19.44$, $p < 0.0001$).

**Experiment 1a: Drift-diffusion models**. To further characterize the processes modulated by tSMS, we fitted drift-diffusion models

to our behavioral data (Fig. 2e). These models assume that the decision process in each trial of the CRT task occurs as a noisy diffusion process that accumulates evidence until either a positive or a negative threshold is reached, respectively, leading to a correct or incorrect response. Error rates and response time distributions for correct and incorrect responses are thus translated into three main parameters of the modeled diffusion process: the drift rate of evidence accumulation $v$, the separation $a$ between decision thresholds, and the non-decision time $T_{er}$ that precedes (and follows) the decision process.

Interestingly, the general increase of initiation times and reduction of errors induced by tSMS compared to sham in the three tasks reflected specific differences in the model parameters in each task (Fig. 2e, f). In the fully-cued task, the threshold

separation was larger after tSMS ($a = 1.05$, 95% confidence interval (c.i.) [1.00–1.12]) compared to sham ($a = 0.92$ [0.88–0.97]; $p = 0.0014$). In the uncued-compatible task, the drift rate was significantly slower after tSMS ($v = 3.74$ [3.57–3.94]) compared to sham ($v = 4.25$ [4.05–4.51]; $p = 0.0023$). Last, in the uncued-incompatible task, the drift rate was significantly faster after tSMS ($v = 3.43$ [3.28–3.60]) compared to sham ($v = 3.14$ [3.01–3.29]; $p = 0.0269$), and the non-decision time significantly longer after tSMS ($T_{er} = 359.9$ [349.8–368.6] ms) compared to sham ($T_{er} = 339.2$ [331.0–346.8] ms; $p = 0.0025$).

**Experiment 1b: Behavioral findings, crossover design.** Sixteen of the subjects who participated in Experiment 1 (8 real, 8 sham) also repeated the experiment at least 1 week later in a double-blind crossover design ($33.1 \pm 8.1$ years old, 11 females). Again, tSMS increased reaction times compared to sham (three-way mixed ANOVA, Stimulation: $F(1,180) = 15.8$, $p = 0.0001$), independently of the task (Stimulation × Task: $F(2,180) = 0.1$, $p = 0.87$) or target location (Stimulation × Location: $F(3,180) < 0.1$, $p > 0.99$; Stimulation × Task × Location: $F(6,180) < 0.1$, $p > 0.99$). The effect was mostly due to increased initiation times (Stimulation: $F(1,180) = 17.6$, $p < 0.0001$), with a smaller but significant increase of movement times (Stimulation: $F(1,180) = 5.8$, $p = 0.0169$). Error rates were particularly low in this subgroup, possibly due to a learning effect in the second session: 2.0% of trials in the fully-cued task, 0.8% in the uncued-compatible task, and 1.4% in the uncued-incompatible task. Consequently, the error rate decrease induced by tSMS did not reach significance ($F(1,180) = 0.16$, $p = 0.69$).

The ability of subjects to correctly guess whether they received real tSMS or sham was not significant after the first session (9 of 16 correct guesses; $\chi^2 = 0.41$, $p = 0.52$), but became significant after second session (13 of 16 correct guesses; $\chi^2 = 6.35$, $p = 0.0117$), with some subjects reporting feeling slower after the real compared to the sham tSMS session. Due to lower error rates and the possible confound of learning effects, drift-diffusion models were not applied to these data.

**Experiment 1c: Behavioral findings, tSMS of M1.** In order to gain insight into possible contribution of SMA-M1 vs. SMA-frontal projections in our behavioral findings, we performed an additional single-blind behavioral experiment in 17 subjects ($29.4 \pm 7.4$ years old, 12 females) who received tSMS over the right M1. The new data were compared against the sham group of Experiment 1a, with no differences in age ($p = 0.47$) or gender ($p = 0.65$).

We found that the ability of M1 tSMS to modulate behavioral performance, as measured by initiation time, was highly dependent on the task (three-way ANOVA, Stimulation × Task, $F(2,436) = 7.4$, $p = 0.0007$). Specifically, M1 tSMS increased the initiation times compared to sham in the fully-cued task (sham: $331.9 \pm 51.4$ ms; real: $395.3 \pm 41.7$ ms; Tukey, $p < 0.0001$), but not in the uncued-compatible nor in the uncued-incompatible tasks ($p > 0.91$). In the fully-cued task, M1 tSMS also significantly decreases the error rate (sham: $4.9 \pm 3.8\%$; real: $2.5 \pm 1.7\%$; unpaired $t$ test, $p = 0.0168$).

At the end of the experiment, 10 of 17 subjects correctly guessed that they had received real tSMS.

**Experiment 2: fMRI findings, local effects.** We tested the functional after-effects of 30-min tSMS of the SMA in a randomized double-blind sham-controlled crossover experiment in 20 subjects ($28.5 \pm 5.2$ years old, 9 females), in which resting-state fMRI was acquired at baseline and immediately after 30 min of tSMS (or sham) applied over the SMA (Fig. 1b). One subject was

discarded due to excessive motion (>35% noisy samples) in at least one acquisition.

We first quantified the local effects induced by tSMS in the cortical area below the magnet, as assessed by changes in ALFF and ReHo in the left and right SMA ROIs defined from the AAL2 atlas. The ALFF was overall higher in the left compared to the right SMA (three-way repeated-measures ANOVA, Side: $F(1,18) = 23.8$, $p = 0.0001$). Importantly, tSMS induced a significant effect compared to sham (Time × Stimulation: $F(1,18) = 5.9$, $p = 0.0260$), which depended on the side of the brain (Time × Stimulation × Side: $F(1,18) = 5.9$, $p = 0.0260$). Namely, tSMS did not induce any detectable change in the left SMA (two-way follow-up ANOVA, Time × Stimulation: $F(1,18) = 1.0$, $p = 0.32$), but its effect was significant in the right SMA (Time × Stimulation: $F(1,18) = 12.2$, $p = 0.0026$), where the ALFF increased after real tSMS (Tukey: $p = 0.0283$) but not after sham ($p = 0.29$). When the time factor was collapsed to analyze ALFF differences (i.e. post–pre stimulation values), in the right SMA the effect of tSMS was significantly greater than sham (two-way repeated-measures ANOVA, Stimulation × Side: $F(1,18) = 5.9$, $p = 0.0260$; Tukey = 0.0005; Fig. 3a).

Comparable results were obtained when analyzing ReHo: (1) higher ReHo in the left compared to the right SMA (three-way repeated-measures ANOVA, Side: $F(1,18) = 13.3$, $p = 0.0018$); (2) significant side-specific effect of tSMS (Time × Stimulation × Side: $F(1,18) = 7.0$, $p = 0.0168$), with no detectable change in the left SMA (two-way follow-up ANOVA, Time × Stimulation: $F(1,18) = 0.0$, $p = 0.89$), and increase in ReHo after tSMS (Time × Stimulation: $F(1,18) = 9.8$, $p = 0.0058$; Tukey: $p = 0.0262$) but not after sham ($p = 0.59$) in the right SMA; (3) greater post–pre effect of tSMS compared to sham in the right SMA (two-way repeated-measures ANOVA, Stimulation × Side: $F(1,18) = 6.9$, $p = 0.0168$; Tukey = 0.0134; Fig. 3b). In fact, ALFF and ReHo measures were correlated both in their baseline values ($r = 0.73$, $p < 0.001$; Fig. 3c) and post–pre differences ($r = 0.55$, $p < 0.001$; Fig. 3d).

At the single-subject level, greater post–pre effect of tSMS compared to sham was observed (i) in 13 of 19 subjects with ALFF, (ii) in 13 of 19 subjects with ReHo, (iii) in 11 subjects with both ALFF and ReHo, and (iv) in 4 subjects with neither ALFF nor ReHo (Fig. 3d).

To control for whether the overall higher local activity we observed in the left compared to the right SMA could be a baseline bias of our dataset, we analyzed resting-state fMRI data from 49 subjects selected at random from the Human Connectome Project (HCP) (age [22–35] years old, 26 females). We found, again, that the ALFF of the left SMA ($0.65 \pm 0.10$) was significantly higher compared to the right SMA ($0.62 \pm 0.11$; paired $t$ test: $p = 0.0004$). This physiological asymmetry might explain the right laterality of tSMS effects on local SMA activity.

**Experiment 2: fMRI findings, distant effects.** We then quantified the distant effects induced by tSMS, as assessed by functional connectivity between the left and right SMA ROIs and the whole brain. Baseline functional connectivity maps of the left and right SMA showed positive connectivity with the expected regions in the motor, salience, and attention networks (Fig. 3e). These positive connectivity maps were used as inclusion masks for the comparison between stimulation conditions (real vs. sham).

Consistent with the right laterality of local effects, distant effects induced by tSMS were significant only when the seed was placed in the right SMA. Specifically, a significant effect of tSMS compared to sham (Time × Stimulation interaction) was observed for a cluster that covered the anterior portion of the left superior temporal gyrus and the lateral portion of the left orbitofrontal

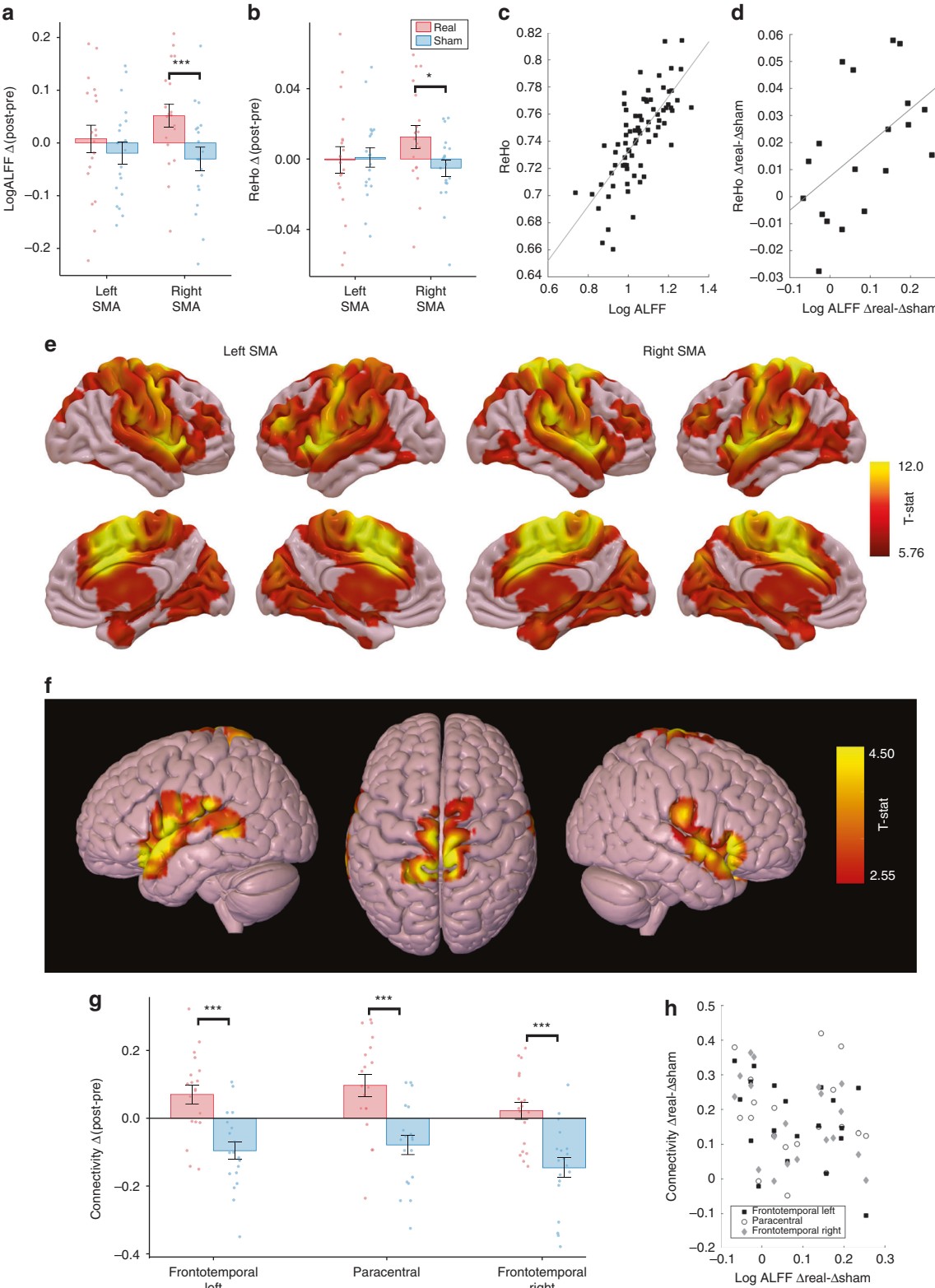

**Fig. 3** Local and distant resting-state fMRI effects induced by tSMS of the SMA. **a**, **b** Average changes induced by tSMS of the SMA on log-transformed ALFF (**a**) and ReHo (**b**) in the left and right SMA. Changes were expressed as delta values (post–pre tSMS or sham). tSMS increased local activity compared to sham in the right SMA. *$P < 0.05$, ***$p < 0.001$. Error bars represent standard errors ($n = 19$). **c**, **d** Correlation between ALFF and ReHo (**c**) in the baseline values ($r = 0.73$, $p < 0.001$; $n = 76$, i.e. 2 sides × 2 sessions × 19 subjects) and (**d**) in the overall effect induced by tSMS vs. sham ($r = 0.55$, $p < 0.001$; $n = 19$; right SMA). **e** Average baseline functional connectivity maps of the left and right SMA. **f** Significant clusters representing the average changes in functional connectivity of the right SMA induced by tSMS compared to sham ($p_{uncorr} < 0.01$, FWE-corrected $p_{cluster} < 0.05$). **g** Corresponding average posthoc changes in the three clusters (***$p < 0.001$, Bonferroni-corrected $t$ test). Error bars represent standard errors ($n = 19$). **h** Individual data of the overall effect induced by tSMS vs. sham on the functional connectivity between the right SMA and the three clusters ($y$-axis) against the overall effect on ALFF ($x$-axis). Data for Fig. 3 are provided at https://osf.io/n3au4/

**Table 1 Regions showing significant increase in functional connectivity with SMA after tSMS compared to sham (Time × Stimulation interaction; $p_{uncorr} < 0.01$, FWE-corrected $p_{cluster} < 0.05$)**

| Cluster | MNI coordinates (mm) | | | | | |
|---|---|---|---|---|---|---|
| | **X** | **Y** | **Z** | **t-peak** | **Cluster extent (voxels)** | **Atlas-based peak location** |
| Left frontotemporal | −40 | 18 | −20 | 5.29 | 2176 | Left OFG/left STG |
| | −38 | −8 | −16 | 5.04 | | Left anterior insula |
| | −56 | 6 | 0 | 4.46 | | Left precentral gyrus |
| Right frontotemporal | 54 | 26 | −4 | 3.88 | 903 | Right IFG |
| | 62 | −12 | 20 | 3.82 | | Right postcentral gyrus |
| | 66 | 6 | −2 | 3.69 | | Right STG |
| Paracentral | 4 | −46 | 78 | 5.29 | 1103 | Postcentral gyrus |
| | −20 | −48 | 64 | 5.07 | | Superior parietal lobe |
| | −2 | −46 | 72 | 4.85 | | Postcentral gyrus |

gyrus, extending to the left anterior insula, the left precentral gyrus in its more ventral–lateral portion, and the inferior frontal gyrus (left frontotemporal cluster; $t$-peak = 5.29; $p_{uncorr} < 0.001$, family-wise error (FWE)-corrected $p_{cluster} < 0.05$). Relaxing the significance threshold ($p_{uncorr} < 0.01$, FWE-corrected $p_{cluster} < 0.05$) uncovered bilateral effects with two additional clusters (Fig. 3f): a cluster mirroring the first cluster on the right side of the brain (right frontotemporal cluster; $t$-peak = 3.88), and another cluster bilaterally located in the dorsal portion of the precentral and postcentral gyri reaching the SMA proper (paracentral cluster; $t$-peak = 5.29). See Table 1 for local maxima of the three clusters. The net effect of tSMS was an increased functional connectivity compared to sham for all clusters (Fig. 3g).

At the single-subject level, greater post–pre effect of tSMS compared to sham was observed in 17 of 19 subjects for each of the three clusters, with 15 subjects showing a net positive effect for all three clusters (Fig. 3h).

In order to investigate whether the increased functional connectivity between the SMA and the distant clusters genuinely reflected increased co-activity of the underlying networks rather than simply being an artifact of the local changes[42], we analyzed the activity in the distant clusters, and we found a trend toward significant change in ALFF (three-way repeated-measures ANOVA, Time × Stimulation: $F(1,18) = 3.7$, $p = 0.069$) and a significant change in ReHo ($F(1,18) = 5.9$, $p = 0.0255$). Namely, with both measures the activity in the distant clusters increased after real tSMS (Fisher: ALFF, $p = 0.0455$; ReHo, $p = 0.0069$) but not after sham (ALFF, $p = 0.56$; ReHo, $p = 0.70$). Furthermore, no positive correlation was observed between local effects and distant effects (Fig. 3h). These results suggest that the distant effects induced by tSMS of the SMA were not simply an indirect consequence of the local effects.

At the end of the experiment, subjects were not able to correctly guess whether they had received real tSMS or sham, neither after the first session (9 of 20 correct guesses; $\chi^2 = 0.20$, $p = 0.65$) nor after the second session (10 of 20 correct guesses; $\chi^2 = 0.00$, $p = 1.00$).

## Discussion

We show that tSMS of the SMA, compared to sham, induces both behavioral and functional after-effects. First, tSMS increases the time to initiate movement while decreasing errors in choice-reaction time tasks, by either elevating decision threshold when withholding predicted actions (fully-cued task), slowing evidence accumulation (i.e. decreasing the drift rate) during motor planning (uncued-compatible task), or increasing non-decision time when solving stimulus–response conflicts (uncued-incompatible

task). Second, tSMS increases the local resting-state fMRI activity of the SMA as well as the bilateral functional connectivity between the SMA and both the paracentral lobule and the frontotemporal cortex, including the inferior frontal gyrus. tSMS over the SMA can thus shift the speed-accuracy tradeoff in favor of accuracy by means of different processes depending of the cognitive demands before overt action, and can modulate both local cortical circuits below the magnet and distant functionally connected cortical networks that may account for specific changes in cognitive processes associated to motor behavior.

**Transcranial static magnetic field stimulation.** tSMS has been recently added to the family of inhibitory NIBS techniques[22]. Previous studies had focused on relatively short applications of tSMS and short-lived after-effects[22–30,32–35,37,39,40,43]. However, we recently showed that tSMS applied for 30 min over M1 induces physiological after-effects that last at least 30 min after the end of the stimulation[31]. Here we thus adopted this new 30-min tSMS protocol, which provided us with sufficient post-stimulation time to measure the after-effects of tSMS using behavioral tasks and resting-state fMRI.

The exact focality of tSMS is unknown, but the magnetic field is not distorted by soft tissue or bone, so its spatial extent only depends on the geometry and physical properties of the magnet[44,45]. By applying tSMS to the SMA, we intended to neuromodulate the entire supplementary motor complex, including both SMA proper and pre-SMA, as confirmed by neuronavigation. We used ROIs from the AAL2 atlas[46] that are consistent with this broad anatomical SMA localization. The rationale was to modulate both more caudal motor-related and more rostral associative-related circuits, thus maximizing the possibility to induce behavioral changes and long-distant fMRI effects, as a first step toward future clinical applications.

The mechanisms of action of tSMS are not completely clear. At the cellular level, tSMS likely alters the function of membrane ion channels due to the diamagnetic anisotropic properties of phospholipids[47–51]. At systems level, 30-min tSMS applied over M1 decreases corticospinal excitability while increasing intracortical excitability[31]. It is not known whether these changes are induced by the sustained exposure to the intensity or to the spatial gradient[52,53] of the magnetic field (e.g. from 2 to 3 cm from the surface along the cylinder axis of the MAG45r there is a magnetic field difference of 74 mT, from 166 to 92 mT, which corresponds to a spatial gradient of 7400 mT/m). Interestingly, the intensity of the static magnetic field inside the scanner is one order of magnitude higher (3 T), but its spatial gradient is virtually zero. Even the fast-switching magnetic gradients applied during MRI acquisitions are two orders of magnitude lower than

tSMS (45 mT/m in our scanner). The fact that we could actually see effects induced by tSMS compared to sham with resting-state fMRI thus suggests that the spatial gradient of the magnetic field may play a mechanistic role in tSMS[52,53]. Importantly, our subjects did not perform the behavioral tasks in the scanner, so any inference between our behavioral and functional findings should be made with caution. Nevertheless, our results suggest that the effects of 30-min tSMS over the SMA reach behavioral relevance and are measureable with resting-state fMRI using a sham-controlled paradigm.

**Behavioral findings**. tSMS of the SMA significantly affected the speed-accuracy tradeoff: initiation times increased in favor of decreasing the error rate. This is consistent with previous studies investigating the SMA role in adjusting behavior. For example, cathodal tDCS of the SMA induces a slowing of reaction times associated with a decrease in the incidence of movement releases by startling acoustic stimuli[54] and results in the prevention of impulsive reactions, reducing the number of errors[55]. Conversely, anodal tDCS of the SMA decreases the probability of withholding anticipated actions[56]. The change in speed-accuracy tradeoff in favor of accuracy is also consistent with the increased efficiency of response suppression observed after pre-SMA stimulation with rTMS inhibitory protocols[57–59]. However, other studies reported the opposite effect[60,61], possibly due to differences among NIBS techniques, tasks, and exact target locations within the SMA. Interestingly, we observed that tSMS of M1 induced a similar increase of initiation times in favor of decreasing errors, but only for the fully-cued task. This suggest that the modulation of the speed-accuracy tradeoff by SMA tSMS in the fully-cued task might be more specifically mediated by SMA-M1 connections, whereas the modulations induced by SMA tSMS in the uncued-compatible and uncued-incompatible tasks might be more specifically mediated by SMA-frontal connections.

Drift-diffusion models allowed us to further characterize the involvement of SMA in choosing adequate behavioral options (the what) and selecting the right moment (the when) and location (the where) of action initiation and execution in behavioral adaptation[62–64]. The increased decision threshold induced by tSMS in the fully-cued task suggests a more cautious, conservative action initiation in deciding when to move while withholding predicted actions. Our result thus extends similar findings of increased caution in deciding where to move, observed with random dot motion tasks after cTBS of the right pre-SMA[65,66] (but see ref. [67]). Interestingly, with the same dot motion tasks cTBS of the dorsolateral prefrontal cortex (DLPFC) was shown to slow down the accumulation of evidence[67], similarly to what we observed with tSMS in the uncued-compatible task, which only involved motor planning. This suggests that both DLPFC and the SMA may be involved in the accumulation of evidence about where to move, depending on the cognitive demands[68]. In addition, the longer non-decision time—and faster accumulation of evidence—induced by tSMS in the uncued-incompatible task offers insight into the cognitive processes underlying the role of the SMA in what to execute in situations of stimulus–response conflict[69–74]. It is noteworthy that non-decision time was markedly longer—and the drift rate slower—in the uncued-incompatible compared to the uncued-compatible tasks, resembling the switch cost in task-switching paradigms[75–77]. Longer non-decision time most likely reflected a delayed decision onset[78], which humans can use as a strategy to trade accuracy over speed as previously shown with motion-interference tasks[79]. The increase of non-decision time induced

by tSMS in the uncued-incompatible task may thus reflect this strategy of delaying decision onset to degrade consolidated/automatic associations for resolving the stimulus–response conflict, thereby allowing evidence to be more efficiently accumulated for launching a well-prepared and more accurate action plan. Overall, the modulations induced by tSMS on the cognitive dynamics that preceded overt behavior in our tasks support a causal role of SMA in determining both when to move in situations of withholding predicted actions, where to initiate and execute the action, and what to execute in situations of stimulus–response conflict.

Drift-diffusion models likely have direct neural correlates. In fact, choice behaviors can be driven by populations of cortical integrator neurons tuned to specific choices, which accumulate information by increasing their firing rate, determining the corresponding behavioral response if a certain threshold is reached[14,80,81]. Considering the decision threshold as the distance between the baseline firing and the firing that determines the behavioral response, the activity of these cortical integration neurons closely corresponds to the parameters of the drift-diffusion model. Therefore, our result that the same perturbation (i.e. tSMS of SMA) modulates different parameters of the drift-diffusion model in different tasks suggests that different subpopulations of neurons may drive when, where, and what decisions in the SMA. This is in agreement with fMRI results suggesting that the decision of what to move is more specifically located in the pre-SMA, whereas the decision of when to move is more confined to the SMA proper[82].

Intriguingly, correctly guessing real tSMS was associated with slower initiation times, and the ability of our subjects to correctly guess the tSMS experimental condition (i.e. real vs. sham) increased after the behavioral experiments (from a tendency after the first session, to a significant guessing ability after the second session), while no guessing ability whatsoever was seen after the fMRI sessions (the experimenter handling tSMS was the same in all cases). Note that no guessing ability was observed in previous studies applying tSMS over other points of the scalp. Two complementary possibilities might explain the present finding. The first one is that during task performance a richer sensorimotor and cognitive feedback (compared to the fMRI session) was perceived via changes in speed and/or errors, a prevailing error awareness function of pre-SMA[83]. The second one is that our stimulation protocol combined with behavioral activation of the local or distant brain circuits modulated the sense of agency of our subjects, that is, the experience of initiating and controlling an action[84], which was previously attributed to SMA activity[85,86]. Even though this finding will require further investigation, it may open new avenues of research in consciousness and action[87].

**fMRI findings**. tSMS of the SMA induced a small but significant increase in the local resting-state fMRI activity, as measured by ALFF and ReHo, particularly in the right SMA. These measures correlate with regional brain metabolism[88–91], suggesting that the overall local effect of tSMS was to increase the metabolism of the SMA. At first glance, this increased metabolism might seem at odds with our behavioral findings, since a higher decision threshold—shifting the speed-accuracy tradeoff towards accuracy—would be expected to reduce the baseline firing of cortical integrator neurons tuned to perceptual choices[14], leading to a decrease rather than increase of pre-SMA activity[92–96]. However, increased regional metabolism is a common observation after the application of inhibitory NIBS techniques, for example, after 1-Hz rTMS[97,98] or cathodal tDCS[99,100], possibly reflecting

increased local levels of inhibitory synaptic activity in specific populations of intracortical neurons. We recently showed that when the same tSMS protocol used here is applied to M1, it does decrease corticospinal excitability, but it also increases intra-cortical facilitation while reducing intracortical inhibition[31]. The increase in local resting-state fMRI activity observed here is thus consistent with tSMS-induced increase of intracortical excit-ability of the SMA, presumably concomitant with reduced baseline activity of specific populations of cortical integrator neurons[14].

Intriguingly, the overall higher local activity we observed in the left compared to the right SMA—both in our dataset and in an independent sample from the HCP—implies that even though the placement of the magnet was symmetric, the baseline functional state of the SMA below the magnet was not. This baseline functional asymmetry is not totally unexpected, since stronger resting-state functional connectivity was previously observed with left compared to right seeds in both pre-SMA[101] and SMA proper[102] (but see ref. [103]). Its causes and implications go beyond the scope of the present work and deserve further investigation, but this asymmetry might explain why the right SMA was more responsive to tSMS than the left SMA in our subjects. The preferential tSMS boost to the right hemisphere, involved in inhibitory control and action switching[13,104–106], might con-tribute to the more cautious action initiation observed in our behavioral experiments.

tSMS of the SMA also increased the functional connectivity between the right SMA and both the paracentral lobule and the frontotemporal cortex bilaterally, including the inferior frontal gyrus. Increased resting-state functional connectivity is not an unusual finding after the application of inhibitory NIBS techniques[107–109]. The increased connectivity with paracentral lobule is in agreement with recent results obtained with 1-Hz rTMS or cTBS of the left SMA[109]. Somewhat surprisingly, however, cTBS of the left SMA seemed to decrease rather than increasing the functional connectivity between the SMA and the inferior frontal gyrus[109]. This difference between the effect of tSMS compared to 1-Hz rTMS or cTBS might be due to mechanistic differences between NIBS techniques, differences in the exact location of stimulation, or to an underlying sham effect (not controlled in ref. [109]). Furthermore, we cannot exclude possible interactions, in terms of metaplasticity[110], between NIBS techniques and the magnetic field of the scanner[25]. These issues will require further investigation for the correct interpretation of resting-state fMRI results in the translation of NIBS techniques to clinical applications.

The increased connectivity between the SMA and the paracentral lobule may explain the motor components of the behavioral modulations induced by tSMS in our tasks, likely subserved by excitatory projections from the SMA proper to M1[3,111,112]. In addition, we cannot exclude contributions from cortico-cortical and cortico-subcortical pathways between the pre-SMA and M1[72,113,114]. Conversely, the increased connec-tivity with the frontotemporal cortex may contribute to the observed shift in speed-accuracy tradeoff, directly mediated by the frontal aslant tract, which is a recently described pathway that connects the SMA with the ventral premotor cortex, including the inferior frontal gyrus (Brodmann areas 44 and 45) and the precentral Brodmann area 6[115–118]. Brodmann areas 44 and 45 are involved in the execution of complex hand movements and sensorimotor integration[119], as part of the movement initiation network[82]. Similarly, the frontal aslant tract also plays a critical role in the execution of visually guided hand movements[120], and the inferior frontal gyrus—particularly on the right side—is a key hub in the network that controls motor inhibition, attentional control and response

switching[106,121,122]. The inferior frontal gyrus also plays a critical role in the accumulation of evidence during action programming[123–126]. Enhanced connectivity between SMA and the frontotemporal clusters could thus contribute to the specific processes used to trade accuracy over speed depending on the cognitive demands in our behavioral tasks. Overall, tSMS of the SMA seems to modulate both more posterior functional networks related to the motor functions of the SMA proper and more frontal networks related the cognitive functions of the pre-SMA.

**Implications for future clinical applications**. tSMS is a portable, easy to apply, inexpensive NIBS technique. The present findings might thus be helpful for informing the use of SMA tSMS for future clinical applications. NIBS of the SMA have provided promising results for treating several brain disorders, such as Tourette syndrome[16–18,127], obsessive compulsive disorder[19], and Parkinson's disease[20,21]. Intriguingly, 1-Hz rTMS of the SMA seems to improve both the cardinal features of Parkinson's dis-ease[21] and levodopa-induced dyskinesias[20]. This dual effect might appear paradoxical from a univocal perspective of local SMA changes, but could be explained by the duality of the main distant pathways modulated by SMA stimulation. On the one hand, the pathway from SMA to M1 might be critical for controlling the cardinal motor features[128]. On the other hand, the pathway from the SMA to the inferior frontal gyrus through the frontal aslant tract might be critical for controlling levodopa-induced dyski-nesias[129]. Beyond Parkinson's disease, the involvement of the frontal aslant tract in both motor function and language, together with the extension of our frontotemporal clusters to the lateral precentral gyrus (i.e. face motor cortex), renders the SMA an appealing target for possible treatment of speech alterations, such as vocal tics, stuttering[130], and aphasia[131,132]. Importantly, the possible differences between our fMRI findings with tSMS of SMA and previous findings with 1-Hz rTMS and cTBS suggests that possible clinical effects might differ—in either positive or negative ways—compared to other inhibitory techniques. The possible relationship between the neurophysiological response to a single tSMS session and the clinical response to repeated ses-sions (see e.g. ref. [133]) will require further investigation. Fur-thermore, the effects of multiple-sessions of tSMS on cortical excitability, connectivity, and clinical symptoms remain to be established.

Overall, our results show that tSMS of the SMA can induce behavioral after-effects associated with the functional modulation of both local cortical circuits below the magnet and distant functionally connected cortical networks. tSMS of SMA may thus be a promising protocol for cognitive research and future clinical applications.

## Methods

**Subjects**. A total of 65 healthy subjects (mean ± SD 30.5 ± 8.2 years old, 43 females) participated in this study. Forty-two subjects participated in Experiment 1 (31.5 ± 8.8 years old, 28 females), 20 subjects in Experiment 2 (28.5 ± 5.2 years old, 9 females), and 10 subjects participating in both. Seventeen subjects (13 had not participated in the previous experiments) participated in the follow-up Experiment 1c. Sample sizes were conservatively determined based on common practice in the relevant literature. Subjects were recruited from our institution and among students from our university. All subjects gave written informed consent. The study was performed according to the Declaration of Helsinki and approved by the local Ethics Committee (Comité Ético de Investigación de HM Hospitales). All experi-ments were performed at CINAC, Hospital Universitario HM Puerta del Sur, Móstoles, Madrid, Spain.

**tSMS protocol**. In all experiments, a cylindrical nickel-plated NdFeB magnet of 45 mm diameter, 30 mm thickness, and 360 g weight (MAG45r; Neurek SL, Toledo, Spain; the Big Magnet in ref. [22]) was used for tSMS, while a non-magnetic steel cylinder, with the same size, weight, and appearance of the magnet, was used for

sham stimulation (MAG45s; Neurek SL, Toledo, Spain). tSMS (or sham) was applied for 30 min with south polarity over the SMA, centered 3 cm anterior to Cz (except in the control Experiment 1c, in which tSMS was applied over the right M1, centered at C4). Subjects were seated comfortably in a semi-darkened room, and were instructed to refrain from speaking and to remain awake while in a calm, relaxed state. At the end of each session (i.e. after the end of the tasks in Experiments 1a–c, and after the post-tSMS fMRI recordings in Experiment 2), subjects were asked to guess whether they received real or sham tSMS. The duration of tSMS application was chosen based on our recent findings showing that 30-min tSMS over M1 reduces corticospinal excitability for at least 30 min after the end of the application[31]. The application of tSMS for 30 min is a safe procedure[134]. In a sample of 10 subjects, we used MRI-guided neuronavigation (Brainsight) to confirm that the target corresponded to the SMA (Fig. 1a). The variability of SMA tSMS targeting was negligible on the mediolateral axis (SD: 1.3 mm) compared to the anteroposterior axis (10.7 mm).

### Experiment 1: Behavioral protocol

Experiment 1a was a randomized double-blind, sham-controlled study with a parallel design (Fig. 1b). Two independent groups of subjects received either tSMS ($n = 20$) or sham ($n = 22$) and, immediately after, performed three CRT tasks (based on ref. [135]). A subgroup of subjects ($n = 16$) also repeated the experiment in a randomized double-blind, sham-controlled crossover design, with at least 1 week between the two experimental sessions (Experiment 1b). Experiment 1c was an additional follow-up experiment in a group of subjects ($n = 17$) who received tSMS over the right M1 instead of SMA in a single-blind design (i.e. subjects were told that they would receive either tSMS or sham, but they all received tSMS).

The CRT tasks consisted of a fully-cued task to assess withholding of predicted actions, an uncued-compatible task to assess motor planning, and an uncued-incompatible task to assess stimulus–response compatibility during motor choice behavior (Fig. 2a). The tasks were performed only after tSMS (or sham) to avoid learning and cumulative effects. In all tasks, two movement parameters were manipulated: hand (right vs. left) and direction (up vs. down), corresponding to four target buttons. Two additional central buttons represented the rest position at the beginning of each trial. For each task, a fixation screen (empty circles) initiated the trial. Subsequently a cue was shown for different inter-stimulus intervals (ISI: 0–200–400–800–1600–3200 ms) with different information quantity in each task. In the fully-cued task, the cue presented one empty circle positioned on top/bottom or right/left position. Subjects were instructed to wait before responding to the corresponding position until the circle filled (go signal). In the uncued-compatible task, the cue showed four empty circles, and therefore showing no prior information of where the movement should be executed. Subjects waited until one filled circle appeared to initiate a movement to the corresponding button. In the uncued-incompatible task, the cue also presented four empty circles, whereby subjects had to wait until one circle filled. However, in this condition, the correct movement was to the button diagonally opposite to that indicated by the filled circle. We calculated four behavioral measures per task and movement position: initiation time (time from the go signal to the release of the home button); movement time (time from the release of the home button till reaching the target button); reaction time (initiation time + movement time); and error rate (percent of incorrectly performed trials). Initiation times, movement times, and reaction times were calculated as the average over correct trials. Each task consisted of a block of 75 trials, with 15 trials at each of the five ISI intervals randomly mixed. An approximately equal right- and left-hand responses were incorporated (counterbalanced). Within each group, the order of the CRT conditions was counterbalanced. The execution of the three tasks lasted 21.5 ± 5.3 min.

### Experiment 1: Drift-diffusion model

Initiation times from Experiment 1a were fitted to drift-diffusion models[41]. When applying drift-diffusion models to our data, we are reducing our CRT tasks to two-choice decisions, with the following assumptions: in the fully-cued task we are modeling the binary go/no-go decision of when to respond (the where is given by the cue); in the uncued-compatible and uncued-incompatible tasks we are modeling a binary choice of correct vs. incorrect movement (e.g. left vs. right hand), respectively without and with stimulus-response conflict. Importantly, although these assumptions of binary decisions limit the interpretation of the model parameters in absolute terms, our interest is not on the absolute values of the parameters but on their differences between tSMS and sham sessions.

Due to the low number of errors, for each task and stimulation group we pooled the data from all subjects into a single population distribution—similarly to previous studies[136–138] after eliminating initiation times >1000 ms and counting initiation times <150 ms as errors (pooled $n = 1400$–1530 trials). Pooling across subjects is justified by the fact that we are not interested in the cognitive mechanisms operating at the level of individual minds, but on the overall effects of tSMS compared to sham at the population level. The resulting population distributions (Fig. 2f) were modeled with the EZ-diffusion model, which analytically estimates decision threshold separation $a$, drift rate of evidence accumulation $v$ and non-decision time $T_{er}$ from the probability of correct responses $P_c$, and the mean and variance of the reaction times (MRT and VRT, respectively),

according to the following equations[139]:

$$v = \text{sign}\left(P_c - \frac{1}{2}\right)s\left\{\frac{\log\left(\frac{P_c}{1-P_c}\right)\left[P_c^2\log\left(\frac{P_c}{1-P_c}\right) - P_c\log\left(\frac{P_c}{1-P_c}\right) + P_c - \frac{1}{2}\right]}{\text{VRT}}\right\}^{\frac{1}{4}} \quad (1)$$

$$a = s^2\log\left(\frac{P_c}{1-P_c}\right)/v, \quad (2)$$

$$\text{MDT} = \left(\frac{a}{2v}\right)\frac{1 - \exp\left(-\frac{va}{s^2}\right)}{1 - \exp\left(-\frac{va}{s^2}\right)}, \quad (3)$$

$$T_{er} = \text{MRT} - \text{MDT}, \quad (4)$$

where MDT is the mean decision time and $s$ is a scaling parameter set at an arbitrary value that represents the standard deviation of the change in the accumulated evidence ($s = 1$ here). Intuitively, the accuracy $P_c$ and the variance of reaction times VRT determine the drift rate $v$, the decision threshold $a$, and the mean decision time MDT. Consequently, the mean reaction time MRT solely contributes to the non-decision time $T_{er}$. Importantly, even though the EZ-diffusion model makes the simplistic assumption of no across-trials variability in the parameters[139], it was suggested to provide unbiased estimates of relative parameter changes between conditions[140] and to be more powerful than the full diffusion model for detecting experimental effects, even on data generated from the full diffusion model[141].

### Experiment 2: fMRI protocol

Experiment 2 was a randomized double-blind, sham-controlled study with a crossover design (Fig. 1c). One group of subjects ($n = 20$) underwent MRI scans before and immediately after tSMS or sham. Each subject underwent two experimental sessions, at least 1 week apart. MRI scans were performed in a 3.0 T scanner (mMR Biograph, Siemens, Erlangen, Germany), with the following acquisition protocol: three-dimensional (3D) T1-weighted magnetization prepared-rapid gradient-echo image with parameters TR/TE/TI 2300/3.34/900 ms, flip angle 8°, and isotropic spatial resolution 1 mm³ (FoV: 256 mm, matrix: 256 × 256, slice thickness: 1 mm); resting-state fMRI using a single-shot gradient-echo planar imaging (EPI) 2D pulse sequence, with acquisition parameters TR/TE 2400/30 ms, optimum flip angle using the Ernst equation (i.e. 79°), isotropic spatial resolution of 3 mm³ (field of view: 192 mm; matrix: 64 × 64; slice thickness: 3 mm), and acceleration factor through parallel imaging × 2 (IPAT2) − the acquisition of this sequence lasted 10 min giving rise to 250 fMRI volumes; and a fieldmap generated from two 2D gradient-echo images, with acquisition parameters TR/TE1/TE2 455/4.92/7.38 ms, flip angle 60°, and same spatial resolution as the fMRI acquisition. During fMRI acquisition, subjects were instructed to remain calm, motionless, without engaging in any cognitive task, and to remain eyes open to avoid falling asleep.

### Experiment 2: Resting-state fMRI pre-processing

T1-weighted MRI images were corrected for intensity bias using N4 bias-correction algorithm[142], denoised using an optimized non-local means filter[143], skull stripped using BET-FSL[144], and segmented in white matter, gray matter, and cerebrospinal fluid (CSF) tissues using FAST-FSL[145]. Skull-stripped images were non-linearly registered to the MNI152 (1 mm³) space using an affine transformation followed by diffeomorphic symmetric normalization in ANTs (Advanced Normalization Tools)[146]. To avoid partial volume contamination from gray matter into the white matter and CSF masks, white matter and CSF masks were eroded. Eroded CSF mask was confined to the ventricles using the mask provided in FSL.

The first five volumes of the resting-state fMRI scan were discarded to allow signal stabilization. Then, the fMRI series were de-spiked, corrected for slice timing, realigned to the first volume, skull stripped, and corrected for EPI distortion using the fieldmap. A scrubbing process was carried out[147], taking three data-quality metrics: frame-wise displacement, the derivative of the root mean square function across voxels (DVARS), and the standard deviation. Both DVARS and standard deviation were computed for motion corrected images using a gray matter mask. Resting-state fMRI volumes overpassing a conservative value of 0.3 for frame-wise displacement, or a cut-off of 75% percentile + 1.5 × inter-quartile range for DVARS and standard deviation were tagged as noisy volumes. Although the noisy samples were not removed from the series during pre-processing, they were not considered for the estimation of functional connectivity[147,148].

The mean of the resting-state fMRI series was subtracted, and linear trends were removed. A linear affine transformation was obtained between the average resting-state fMRI volume and the T1-weighted image using ANTS. The inverse of this transformation was applied to the white matter and ventricle CSF masks in order to extract white matter and ventricle CSF average resting-state fMRI activity. The nuisance regressors matrix contained the white matter and ventricle CSF average time-series, the six motion-correction residuals, and their derivatives. All 16 regressors were Z-scored, not considering the noisy volumes, and were regressed out from the dataset. The resulting residuals were band-pass filtered (0.009–0.08 Hz) using a second-order Butterworth filter. Pre-processed resting-state fMRI was normalized to MNI152 (3 mm³) and smoothed using a 8-mm full-width at half-maximum Gaussian kernel.

**Experiment 2: Resting-state fMRI analysis**. Left and right SMA ROIs were extracted from the AAL2 atlas[46]. For each ROI, we estimated the following measures: the power of the resting-state fMRI time-series in the selected frequency band, also known as ALFFs, computed as the variance of the time-series without considering the noisy samples; the ReHo, which quantifies the short-range temporal similarity by computing the Kendall's coefficient of concordance[149]; and the functional connectivity maps, computed as the Z-transformed Pearson's linear correlation between the resting fMRI time-series of each ROI and all voxels in the whole brain.

**HCP data**. We also analyzed resting-state fMRI from 49 subjects from the 2014 HCP S1200 data release, which is publicly available at http://humanconnectome.org (see ref. [150] for a detailed explanation of the entire acquisition protocol). Briefly, HCP MRI data were acquired on a 3 T Siemens Skyra-Connectome scanner. For resting-state fMRI acquisitions, the participants were with their eyes open and fixated on a cross hair on the screen. A simultaneous multi-slice pulse sequence with an acceleration factor of eight was used to acquire two resting-state fMRI runs with opposite phase encoding direction (right-left (RL) and left-right (LR)). Each run consisted of 1200 volumes with parameters TR/TE (repetition time/echo time) 720/33.1 ms, flip angle 52°, and isotropic spatial resolution 2-mm isotropic spatial resolution. Only the acquisition with LR phase encoding direction was used in here. We used the pre-processed and artifact-removed resting-state fMRI series as provided by the HCP S1200 data release[151]. Pre-processed series were temporally demeaned, linearly detrended, and band-pass filtered (0.009–0.08 Hz) using a second-order Butterworth filter. The ALFF was computed as the variance of the time-series in the left and right SMA ROI, as in Experiment 2, but without tagging and discarding noisy samples.

**Statistics and reproducibility**. In Experiment 1a (parallel design), reaction times, initiation times, movement times, and error rates were entered separately into three-way independent-measures ANOVA, with the following factors: Task (fully-cued, uncued-compatible, and uncued-incompatible), Location (upper left, lower left, upper right, lower right), and Stimulation (real, sham). Two-way follow-up ANOVAs were applied for individual tasks in case of significant interaction. The analyses were also repeated by adding a fourth factor—Guessing (correct, incorrect)—representing the ability of subjects to guess whether they had received real tSMS or sham, when asked at the end of the experiment. The same ANOVA strategy was used in Experiment 1b (crossover design, the Stimulation factor was compared within subjects) and Experiment 1c (using the sham data of Experiment 1a). The Pearson's $\chi^2$ test was used to evaluate the a posteriori ability of subjects to discriminate whether they had received real or sham stimulation. Results were considered significant at $p < 0.05$.

For the drift-diffusion model analysis, we employed bootstrapping techniques for constructing 95% confidence intervals on the estimated parameters and for performing hypothesis testing on the effects of tSMS compared to sham. Namely, for each task and stimulation condition, to construct 95% confidence we obtained 100,000 bootstrapped datasets by random resampling with replacement from the measured initiation times *within* stimulation conditions, then we applied the EZ-diffusion model to each bootstrapped dataset, and we extracted the 2.5% and 97.5% percentile from the distribution of the estimated parameters. For performing hypothesis testing, we obtained 100,000 bootstrapped pairs of datasets by random resampling with replacement from the initiation times *between* stimulation conditions (i.e. from the entire dataset of both tSMS and sham groups together), we applied the EZ-diffusion model separately to each pair of bootstrapped datasets, and for each parameter of the model ($a$, $v$, and $T_{er}$), we estimated the $p$ value as the probability that the absolute difference between the bootstrapped pair of datasets was greater than the absolute difference between the actual tSMS-sham values. The estimated $p$ value represents the probability that the model parameters estimated from the tSMS and sham groups are drawn from the same statistical distribution of initiation times (null hypothesis), or from different distributions (alternative hypothesis). Since we had three tasks, we applied Bonferroni correction for multiple comparisons (multiplying $p$ values by 3). Results were considered significant at $p < 0.05$.

In Experiment 2, ALFF (log-transformed) and ReHo were separately entered into three-way repeated-measures ANOVA, with Time (pre, post stimulation), Stimulation (real, sham), and Side (left, right ROI) as main factors, followed by follow-up ANOVAs for each Stimulation condition in case of significant interaction and Tukey's post hoc tests. Post–pre stimulation values were also separately entered into two-way repeated-measures ANOVA, followed by Tukey's post hoc tests. Results were considered significant at $p < 0.05$. Group functional connectivity maps for left and right SMA were separately obtained by running a random-effect second-level analysis in SPM12, using a one-sample $t$ test. Statistical maps showing group connectivity were thresholded at the voxel level ($p < 0.05$, FWE-corrected for multiple comparisons) to define inclusion masks of positive baseline functional connectivity prior to the statistical comparison between real and sham tSMS. The effect of tSMS was separately tested for left and right SMA functional connectivity by considering the Time × Stimulation interaction significant at $p_{uncorr} < 0.001$, with FWE correction at the cluster level ($p_{cluster} < 0.05$). ALFF (log-transformed) and ReHo were also calculated in the distant clusters of significant functional connectivity and separately entered into three-way

repeated-measures ANOVA, with Time (pre, post stimulation), Stimulation (real, sham), and Cluster as main factors, followed by Fisher's least significant difference post hoc tests. Results were considered significant at $p < 0.05$. Again, the Pearson's $\chi^2$ test was used to evaluate the a posteriori ability of subjects to discriminate whether they had received real or sham stimulation.

In the Human Connectome Project Data, left vs. right ALFF values were compared with a two-sided paired $t$ test.

**Reporting summary**. Further information on research design is available in the Nature Research Reporting Summary linked to this article.

## Data availability
The datasets generated during and/or analyzed during the current study are available in the OSF repository, https://osf.io/n3au4/.

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

## Acknowledgements

This work was supported by MINECO/AEI/FEDER-UE (SAF2016-80647-R and SAF2017-86246-R) and The Michael J. Fox Foundation (Grant 9205). J.A.P.-P. was supported by the Spanish Ministry of Education through the National Program Juan de la Cierva (FJCI-2015-25095). I.O. was supported by the Carlos III Health Institute with a post-doctoral grant (Sara Borrell, CD15/00092). Data were provided in part by the Human Connectome Project, WU-Minn Consortium (Principal Investigators: David Van Essen and Kamil Ugurbil; 1U54MH091657) funded by the 16 NIH Institutes and Centers that support the NIH Blueprint for Neuroscience Research, and by the McDonnell Center for Systems Neuroscience at Washington University.

## Author contributions

J.A.P.-P., I.O., A.O., and G.F. designed the study. J.A.P.-P., I.O., and G.F. performed the behavioral and imaging experiments. P.G. contributed to behavioral experiments and analyses. J.A.P.-P. performed imaging analysis. G.F. performed statistical analysis. J.A.P-P and G.F. prepared the figures. J.A.P.-P., I.O., and G.F. drafted the first version of the manuscript. J.A.P.-P., I.O., P.G., M.D., B.A.S., J.A.O., A.O., and G.F. provided critical review and edited the manuscript.

## Competing interests

A.O. and G.F. declare the following competing interests: are cofounders of the company Neurek SL, which is a spinoff of the Foundation of the Hospital Nacional de Parapléjicos, and inventors listed on the following patents: P201030610 and PCT/ES2011/070290 (patent abandoned). J.A.P.-P., I.O., P.G., M.D., B.A.S., and J.A.O. declare no competing financial or non-financial interests.
