## [Peer Review File · Communications Biology]

Reviewers' comments:

Reviewer #1 (Remarks to the Author):

The authors examined the effect of static magnetic field stimulation (tSMS) over the supplementary motor area (SMA) in healthy subjects. As results, at behavioral level, tSMS over the SMA induce behavioral aftereffects associated with modulation of both local and distinct functionally-connected cortical circuits involved in the control of speed-accuracy tradeoff. These results are interesting and well-written and the effect of tSMS on not only neurophysiological aspect not only behavioral level is clearly presented and more straightforward. Also, this study is important for application of NIBS including tSMS. I have listed only one questions.

Numerous studies have reported NIBS have higher "inter-individual variability". In regards to tSMS over SMA, were there inter-individual variability? Considering application for rehabilitation, I would you like to present individual data in this paper. If possible, I would like to describe the characteristics of responder for tSMS over SMA.

Reviewer #2 (Remarks to the Author):

This paper reported behavioral and fMRI impact of tSMS over the SMA. The main findings were: 1) tSMS changed the speed-accuracy tradeoff towards accuracy, i.e. less errors at the cost of slower reaction time, and 2) it modulated both local activity and network connecting distant structures. The method is novel and the results are relevant in the context of motor neuroscience (and potentially clinical application of NIBS).

A few control experiments, if not very large in terms of sample-size, would greatly strengthen the author's conclusion and possibly shorten part of the discussion where several possibilities are listed as the potential mechanisms of the observed findings.

1) If we had several control stimulation sites in addition to the current one, some of the issues including laterality and difference between SMA and pre-SMA would have been clearer. Do the authors have any comments on this from the neuronavigation results? And is it possible to add some additional data using different stimulation sites?

2) Effect of tSMS seems very variable and specific with respect to the task: it affected decision threshold in the fully-cued task, evidence accumulation in the uncued-compatible task, and non-decision time in the uncued-incompatible task. What is the cause/mechanism of such difference? I can understand that the SMA plays multiple roles as described in page 12, but I could not see logical explanation of the connections between the task and induced change by tSMS (e.g. why not decision threshold, but evidence accumulation, for the uncued-compatible task?).

3) Blinding of the participants was successful in the fMRI experiment but failed in the behavioral task. Performing the task, as the authors claim, is a plausible one. But, given that tSMS was delivered before the task performance, how did the "richer sensorimotor and cognitive feedback" exert substantial influence of the guessing of real/sham?

Reviewer #3 (Remarks to the Author):

Review on the paper:

"Transcranial static magnetic field stimulation of the supplementary motor area modulates local

resting-state activity, distant functional connectivity and motor behavior”

The paper is very interesting and timely providing new information about the function of supplementary motor areas. The method (tSMS) is new to me, but I find enough background and references to method. It is also closely linked to future clinical applications and the aim of the study is justified. The methods seem sound and the study is statistically well powered. To my understanding, the statistics is performed correctly, sometimes with several thresholds of significance. Scientific quality is high, and technically, it is well planned and performed. The results are shown in transparent way. The text is clear and easy to follow, and the results are interpreted carefully. References are numerous and performed by several independent groups when applicable. The findings are discussed in relation to other NIBS methods. As such, I only find minor remarks.

Specific points:

- 1) What does 'but' refer to? (page 3, line 59). This could be opened up some more.
- 2) A few remarks about statistics: What does two proportion test refer to (page 5, line 86)? What does Fisher refer to? (page 9, line 228).
- 3) What were the reasons that not all subjects completed all tasks? (Page 5, lines 86-88).
- 4) Reference Ratcliff et al., 2016 does not need to be mentioned in results (page 6, line 117) since it is explained in the methods.
- 5) Subscript in T_{er} is not clear in the results compared to methods.
- 6) In the beginning of the discussion, there is an expression 'slowing evidence accumulation' which is a bit unclear to me.
- 7) There are a few spelling mistakes: Nevertheless (page 11, line 283), DLFP (page 12, line 314)
- 8) Can you very briefly add the reason for observed increased regional metabolism after the application of 'inhibitory' NIBS techniques (page 14, line 359)? I also would like to have more information of cortical integrator neurons though there is a reference given (page 14, line 366).
- 9) The Brodmann areas are only referred as areas. (e.g. page 15, line 410).
- 10) Is SMA target meant to be 3 cm anterior (as in the text) or above to Cz? (Figure 1). In the same figure, behavior could be specified some more, or refer to next figure.
- 11) Legend, Figure 2, should there be v instead of o? (page 37, line 1044)

Reviewer #1 (Remarks to the Author):

The authors examined the effect of static magnetic field stimulation (tSMS) over the supplementary motor area (SMA) in healthy subjects. As results, at behavioral level, tSMS over the SMA induce behavioral aftereffects associated with modulation of both local and distinct functionally-connected cortical circuits involved in the control of speed-accuracy tradeoff. These results are interesting and well-written and the effect of tSMS on not only neurophysiological aspect not only behavioral level is clearly presented and more straightforward. Also, this study is important for application of NIBS including tSMS. I have listed only one questions. Numerous studies have reported NIBS have higher “inter-individual variability”. In regards to tSMS over SMA, were there inter-individual variability? Considering application for rehabilitation, I would you like to present individual data in this paper. If possible, I would like to describe the characteristics of responder for tSMS over SMA.

RESPONSE

We thank the reviewer for the positive feedback. We agree that inter-individual variability is an important issue, so we expanded on this point in the revised manuscript. Our behavioral experiments were not designed to assess inter-individual variability in the response to SMA tSMS, but our neuroimaging experiment may offer some insight.

Fig. 3D in the original manuscript represents the individual data of the effects induced by tSMS on ALFF and ReHo in the right SMA. From the figure it can be seen that (i) a positive effect of tSMS compared to sham was observed on ALFF in 13 of 19 subjects, (ii) on ReHo on 13 of 19 subjects, (iii) on *both* ALFF and ReHo in 11 subjects, (iv) neither on ALFF nor ReHo in 4 subjects. These numbers were clarified in the results section of the revised manuscript.

Lines 217-219

“At the single-subject level, greater post-less-pre effect of tSMS compared to sham was observed (i) in 13 of 19 subjects with ALFF, (ii) in 13 of 19 subjects with ReHo, (iii) in 11 subjects with both ALFF and ReHo, (iv) in 4 subject with neither ALFF nor ReHo (Fig. 3D).”

We also added a new Fig. 3H with the individual data of the effects induced by tSMS on the functional connectivity between the right SMA and the three significant clusters.

A positive effect of tSMS compared to sham was observed in 17 of 19 subjects

for each of three clusters, with 15 subjects showing a positive effect for all three clusters. No positive correlation was observed between local effects and distant effects. This was clarified in the results of the revised manuscript.

Lines 250-252

“At the single-subject level, greater post-less-pre effect of tSMS compared to sham was observed in 17 of 19 subjects for each of the three clusters, with 15 subjects showing a net positive effect for all three clusters (Fig. 3H).”

The possible relationship between the neurophysiological response to a single-session and the clinical response to repeated sessions, hinted at by the reviewer, is intriguing (see e.g. Oliveira-Maia et al., Brain Stimul 2017) and will require further investigation. This was clarified in the discussion.

Lines 490-492

“The possible relationship between the neurophysiological response to a single tSMS session and the clinical response to repeated sessions (see e.g. Oliveira-Maia et al., 2017) will require further investigation.”

Reviewer #2 (Remarks to the Author):

This paper reported behavioral and fMRI impact of tSMS over the SMA. The main findings were: 1) tSMS changed the speed-accuracy tradeoff towards accuracy, i.e. less errors at the cost of slower reaction time, and 2) it modulated both local activity and network connecting distant structures. The method is novel and the results are relevant in the context of motor neuroscience (and potentially clinical application of NIBS).

A few control experiments, if not very large in terms of sample-size, would greatly strengthen the author's conclusion and possibly shorten part of the discussion where several possibilities are listed as the potential mechanisms of the observed findings.

RESPONSE

We thank the reviewer for the positive feedback. We added a control experiment to gain mechanistic insight, as described in response to point 1 below.

1) If we had several control stimulation sites in addition to the current one, some of the issues including laterality and difference between SMA and pre-SMA would have been clearer. Do the authors have any comments on this from the neuronavigation results? And is it possible to add some additional data using different stimulation sites?

RESPONSE

The neuronavigation data indicate that the variability of the tSMS target was negligible on the mediolateral axis (SD: 1.3 mm) compared to the anteroposterior axis (10.7 mm). This was clarified in methods. It is thus unlikely for the laterality of tSMS effects to be due to a laterality bias of tSMS targeting.

Lines 528-529

“The variability of SMA tSMS targeting was negligible on the mediolateral axis (SD: 1.3 mm) compared to the anteroposterior axis (10.7 mm).”

We agree with the reviewer that testing other stimulation sites might clarify some issues. However, each new stimulation site would also represent an entirely new study with its own new issues.

As a compromise, in the revised manuscript we added a new behavioral experiment in 17 subjects who received tSMS over the right motor cortex (M1). The new experiment was single blind, i.e. subjects were told that they would receive either tSMS or sham, but they all received tSMS; the new data (Experiment 1c) were compared against the sham group of Experiment 1a. The rationale for the new stimulation site is threefold: (i) right M1 is the only target for which we know the neurophysiological effects of 30-min tSMS (published in our recent study, Dileone et al., Brain Stimul 2018); (ii) the right hemisphere was more sensitive to SMA tSMS in the fMRI experiment presented in the original manuscript; (iii) M1 tSMS might allow us to partly dissociate the contribution of SMA-M1 connections (compared to SMA-frontal connections) to the behavioral effects we observed after SMA tSMS.

Interestingly, we found that the ability of M1 tSMS to modulate behavioral performance, as measured by initiation time, was highly dependent on the task (three-way ANOVA, real-sham x task, $F(2,436)=7.4$, $p=0.0007$). Specifically, M1 tSMS increased the initiation times compared to sham in the fully-cued (FC) task (Tukey, $p<0.0001$), but not in the uncued-compatible (UC) nor in uncued-incompatible (UI) tasks ($p>0.91$). In the FC task, M1 tSMS also significantly decrease the error rate in the FC task (unpaired t-test, $p=0.0168$). These results suggest that the modulation of the speed-accuracy tradeoff by SMA tSMS in the FC task might be more specifically mediated by SMA-M1 connections, whereas the modulations induced by SMA tSMS in the UC and UI tasks might be more specifically mediated by SMA-frontal connections. This was added in the methods, results and discussion of the revised manuscript.

Lines 172-185

“In order to gain insight into possible contribution of SMA-motor cortex vs SMA-frontal projections in our behavioral findings, we performed an additional single-blind behavioral experiment in 17 subjects (29.4 ± 7.4 years old, 12 females) who received tSMS over the right motor cortex. The new data were compared against the sham group of Experiment 1a, with no differences in age ($p=0.47$) or gender ($p=0.65$).

We found that the ability of motor cortex tSMS to modulate behavioral performance, as measured by initiation time, was highly dependent on the task (three-way ANOVA, real-sham x task, $F(2,436)=7.4$, $p=0.0007$). Specifically, motor cortex tSMS increased the initiation times compared to sham in the fully-cued (FC) task (sham: 331.9 ± 51.4 ms; real: 395.3 ± 41.7 ms; Tukey, $p<0.0001$), but not in the uncued-compatible (UC) nor in uncued-incompatible (UI) tasks ($p>0.91$). In the FC task, M1 tSMS also significantly decrease the error rate in the FC task (sham: 4.9 ± 3.8 ms; real: 2.5 ± 1.7 ms; unpaired t-test, $p=0.0168$).

At the end of the experiment, 10 of 17 subjects correctly guessed that they had received real tSMS.”

Lines 332-338

“Interestingly, we observed that tSMS of motor cortex induced a similar increase of initiation times in favor of decreasing the error rates, but only for the fully-cued task. This suggest that the modulation of the speed-accuracy tradeoff by SMA tSMS in the fully-cued task might be more specifically mediated by SMA-motor cortex connections, whereas the modulations induced by SMA-tSMS in the uncued-compatible and uncued-incompatible tasks might be more specifically mediated by SMA-frontal connections.”

Lines 547-549

“Experiment 1c was an additional follow-up experiment in a group of subjects ($n=17$) who received tSMS over the right motor cortex instead of SMA in a single-blind design (i.e. subjects were told that they would receive either tSMS or sham, but they all received tSMS).”

2) Effect of tSMS seems very variable and specific with respect to the task:

it affected decision threshold in the fully-cued task, evidence accumulation in the uncued-compatible task, and non-decision time in the uncued-incompatible task. What is the cause/mechanism of such difference? I can understand that the SMA plays multiple roles as described in page 12, but I could not see logical explanation of the connections between the task and induced change by tSMS (e.g. why not decision threshold, but evidence accumulation, for the uncued-compatible task?).

RESPONSE

The reviewer is correct in that the effects of tSMS, when analyzed with drift-diffusion models, were task specific. This was not unexpected, since the tasks were designed to engage different decision processes during behavior. Drift-diffusion models of behavioral responses likely have direct neural correlates. In fact, choice behaviors can be driven by populations of cortical integrator neurons tuned to specific choices, which accumulate information by increasing their firing rate, determining the corresponding behavioral response if a certain threshold is reached (Roitman and Shadlen, J Neurosci 2002; Mazurek et al., Cereb Cortex 2003; Bogacz et al., Trends Neurosci 2010). Considering the decision threshold as the “distance” between the baseline firing and the firing that determines the behavioral response, the activity of these cortical integration neurons closely correspond to the parameters of the drift-diffusion model. Therefore, our results that the same perturbation (i.e. tSMS of SMA) modulates different parameters of the drift-diffusion model in different tasks suggest that different subpopulations of neurons may drive “when”, “where” and “what” decisions in the SMA. This was clarified in the discussion.

Lines 375-384

“Drift-diffusion models likely have direct neural correlates. In fact, choice behaviors can be driven by populations of cortical integrator neurons tuned to specific choices, which accumulate information by increasing their firing rate, determining the corresponding behavioral response if a certain threshold is reached (Roitman and Shadlen, 2002; Mazurek et al., 2003; Bogacz et al., 2010). Considering the decision threshold as the “distance” between the baseline firing and the firing that determines the behavioral response, the activity of these cortical integration neurons closely correspond to the parameters of the drift-diffusion model. Therefore, our result that the same perturbation (i.e. tSMS of SMA) modulates different parameters of the drift-diffusion model in different tasks suggests that different subpopulations of neurons may drive “when”, “where” and “what” decisions in the SMA.”

3) Blinding of the participants was successful in the fMRI experiment but failed in the behavioral task. Performing the task, as the authors claim, is a plausible one. But, given that tSMS was delivered before the task performance, how did the “richer sensorimotor and cognitive feedback” exert substantial influence of the guessing of real/sham?

RESPONSE

Sorry for not being clear about this methodological point: tSMS was indeed delivered before the task performance, but it is after the tasks when subjects

were asked to guess whether they received real or sham tSMS. This was clarified in the methods of the revised manuscript.

Lines 521-523

At the end of each session (i.e. after the end of the tasks in experiments 1a-c, and after the post-tSMS fMRI recordings in experiment 2), subjects were asked to guess whether they received real or sham tSMS.

Reviewer #3 (Remarks to the Author):

Review on the paper:

"Transcranial static magnetic field stimulation of the supplementary motor area modulates local resting-state activity, distant functional connectivity and motor behavior"

The paper is very interesting and timely providing new information about the function of supplementary motor areas. The method (tSMS) is new to me, but I find enough background and references to method. It is also closely linked to future clinical applications and the aim of the study is justified. The methods seem sound and the study is statistically well powered. To my understanding, the statistics is performed correctly, sometimes with several thresholds of significance. Scientific quality is high, and technically, it is well planned and performed. The results are shown in transparent way. The text is clear and easy to follow, and the results are interpreted carefully. References are numerous and performed by several independent groups when applicable. The findings are discussed in relation to other NIBS methods. As such, I only find minor remarks.

RESPONSE

We are thankful to the reviewer for the positive feedback.

Specific points:

1) What does 'but' refer to? (page 3, line 59). This could be opened up some more.

RESPONSE

In the revised manuscript, we clarified that "Despite one study reporting negative findings (Kufner et al., 2017), possibly due to methodological differences (Foffani and Dileone, 2017), converging evidence supports the ability of tSMS to induce local effects."

2) A few remarks about statistics: What does two proportion test refer to (page 5, line 86)? What does Fisher refer to? (page 9, line 228).

RESPONSE

They refer to "two proportion z-test" and "Fisher least-significance test". This was clarified in the results (two proportion) and methods (Fisher) of the revised manuscript.

3) What were the reasons that not all subjects completed all tasks? (Page 5, lines 86-88).

RESPONSE

Two subjects did not follow correctly the instructions in one task. This was clarified in the results.

4) Reference Ratcliff et al., 2016 does not need to be mentioned in results (page 6, line 117) since it is explained in the methods.

RESPONSE

The reference was omitted from the results.

5) Subscript in Ter is not clear in the results compared to methods.

RESPONSE

Corrected

6) In the beginning of the discussion, there is an expression ‘slowing evidence accumulation’ which is a bit unclear to me.

RESPONSE

We clarified that this refers to “decreasing the drift rate”.

7) There are a few spelling mistakes: Nevertheless (page 11, line 283), DLFP (page 12, line 314)

RESPONSE

Corrected

8) Can you very briefly add the reason for observed increased regional metabolism after the application of ‘inhibitory’ NIBS techniques (page 14, line 359)? I also would like to have more information of cortical integrator neurons though there is a reference given (page 14, line 366).

RESPONSE

We clarified that the increased regional metabolism after the application of ‘inhibitory’ NIBS techniques may reflect “increased local levels of inhibitory synaptic activity in specific populations of intracortical neurons”.

About the cortical integrator neurons, we added a brief paragraph in the behavioral part of the discussion:

“Drift-diffusion models of behavioral responses likely have direct neural correlates. In fact, choice behaviors can be driven by populations of cortical integrator neurons tuned to specific choices, which accumulate information by increasing their firing rate, determining the corresponding behavioral response if a certain threshold is reached (Roitman and Shadlen, 2002; Mazurek et al., 2003; Bogacz et al., 2010). Considering the decision threshold as the “distance” between the baseline firing and the firing that determines the behavioral response, the activity of these cortical integration neurons closely correspond to the parameters of the drift-diffusion model. Therefore, our results that the same perturbation (i.e. tSMS of SMA) modulates different parameters of the drift-diffusion model in different tasks suggest that different subpopulations of neurons may drive “when”, “where” and “what” decisions in the SMA.”

9) The Brodmann areas are only referred as areas. (e.g. page 15, line 410).

RESPONSE

Corrected

10) Is SMA target meant to be 3 cm anterior (as in the text) or above to Cz? (Figure 1). In the same figure, behavior could be specified some more, or refer to next figure.

RESPONSE

The SMA target was 3 cm anterior to Cz. We corrected the legend. In the same legend we now specify that “behavior” means choice-reaction time tasks and refer to Fig. 2.

11) Legend, Figure 2, should there be v instead of o? (page 37, line 1044)

RESPONSE

“o” should be “or”, while “0” (i.e. zero) is correct. This was corrected in the legend.

REVIEWERS' COMMENTS:

Reviewer #1 (Remarks to the Author):

The manuscript has been revised well.
I think this manuscript will be acceptable for this journal.

Reviewer #2 (Remarks to the Author):

I thank the authors for the comprehensive revision.
Regarding my comment #2, could the authors please clarify which subpopulation(s) of SMA should correspond to "when," which to "where" and so forth. Without this clarification, it seems hard to see logical explanation of the connections between the task and induced change by tSMS.

Reviewer #3 (Remarks to the Author):

I am satisfied with the response.

Reviewer 2

I thank the authors for the comprehensive revision. Regarding my comment #2, could the authors please clarify which subpopulation(s) of SMA should correspond to "when," which to "where" and so forth. Without this clarification, it seems hard to see logical explanation of the connections between the task and induced change by tSMS.

RESPONSE

In response to the reviewer's request, we added the following clarification in the discussion of the revised manuscript: "This is in agreement with fMRI results suggesting that the decision of what to move is more specifically located in the pre-SMA, whereas the decision of when to move is more confined to the SMA-proper⁸²."